# A Novel m7G-Related Genes-Based Signature with Prognostic Value and Predictive Ability to Select Patients Responsive to Personalized Treatment Strategies in Bladder Cancer

**DOI:** 10.3390/cancers14215346

**Published:** 2022-10-29

**Authors:** Guichuan Lai, Xiaoni Zhong, Hui Liu, Jielian Deng, Kangjie Li, Biao Xie

**Affiliations:** Department of Epidemiology and Health Statistics, School of Public Health, Chongqing Medical University, Yixue Road, Chongqing 400016, China

**Keywords:** N7-methylguanosine, prognosis, tumor microenvironment, GSVA, immunotherapy, chemotherapeutic drugs, predict, biomarker, bioinformatics

## Abstract

**Simple Summary:**

Although N7-methylguanosine (m7G) modification serves as a tumor promoter in bladder cancer (BLCA), the comprehensive role of m7G-related characterization in BLCA remains unclear. In this study, we integrated three cohorts consisting of 760 BLCA patients and used a consensus unsupervised clustering method to classify the BLCA patients into different clusters based on 18 m7G-related genes. Next, we identified some candidate proteins via various approaches to reveal the potential mechanism among these m7G-related clusters. Finally, we developed a novel scoring system depending on the GSVA algorithm. The constructed m7G-related signature can accurately be used for risk stratification. Most importantly, our signature can effectively predict patients’ response to immunotherapy and some targeted chemotherapeutic drugs. We believe our research can guide more clinical therapeutic strategies, improving the survival of BLCA patients in the future.

**Abstract:**

Although N7-methylguanosine (m7G) modification serves as a tumor promoter in bladder cancer (BLCA), the comprehensive role of m7G-related characterization in BLCA remains unclear. In this study, we systematically evaluated the m7G-related clusters of 760 BLCA patients through consensus unsupervised clustering analysis. Next, we investigated the underlying m7G-related genes among these m7G-related clusters. Univariate Cox and LASSO regressions were used for screening out prognostic genes and for reducing the dimension, respectively. Finally, we developed a novel m7G-related scoring system via the GSVA algorithm. The correlation between tumor microenvironment, prediction of personalized therapies and this m7G-related signature was gradually revealed. We first identified three m7G-related clusters and 1108 differentially expressed genes relevant to the three clusters. Based on the profile of 1108 genes, we divided BLCA patients into two clusters, which were quantified by our established m7G-related scoring system. Patients with higher m7G-related scores tended to have a better OS and more chances to benefit from immunotherapy. A significantly negative connection between sensitivity to classic chemotherapeutic drugs and m7G-related signature was uncovered. In summary, our data show that m7G-related characterization of BLCA patients can be of value for prognostic stratification and for patient-oriented therapeutic options, designing personalized treatment strategies in the preclinical setting.

## 1. Introduction

Bladder cancer (BLCA) is the most common malignant tumor of the urinary system, with high incidence and mortality globally [1]. Existing treatments mainly include cisplatin-based chemotherapy, immunotherapy, intravesical treatment and neoadjuvant chemotherapy [2]. Although these therapies can improve the prognosis of the disease, a number of patients show a low or no response to treatments, partly because of high tumor heterogeneity [3]. With the development of more advanced methodologies and technologies, molecular subtyping of BLCA cases has provided some clues for selecting upfront, before starting treatment, which patients will respond to the appropriate therapies [4]. Recently, novel approaches based on bioinformatics and machine learning have attracted attention to the possibility of predicting prognosis and selecting appropriate therapies [5,6,7]. There is still a need, however, to develop novel genetic signatures with clinical value, in order to transfer the high amount of data obtained with the bioinformatic approach to the clinical setting [8].

N7-methylguanosine (m7G) is defined as a modification, where the seventh N position of RNA guanine (G) is complexed with a methyl group through the action of methylation transferase [9]. As a new RNA modification, m7G mediates various vital biological processes by adding m7G modification to target RNAs, leading to the impact on the production, structure and maturation of RNA [10]. This modification mainly occurs in tRNA, mRNA, ribosomal RNA, small nuclear RNA and small nucleolar RNA [11]. Moreover, this modification plays the role of a double-edged sword in the progression of tumors [12]. For example, METTL1 and WDR4 serve as tumor promoters, resulting in tumor growth, invasion and metastasis in hepatocarcinoma, esophageal squamous cell carcinoma, glioma, head and neck squamous cell carcinoma and non-small cell lung cancer [13,14,15,16,17,18]. However, METTL1 exerts a strong anti-cancer effect on colon cancer [19]. In BLCA, METTL1 and WDR4 favor the development of BLCA through the regulation of m7G [20]. METTL1 is highly expressed in BLCA tissues, and its expression is positively correlated with a poor prognosis of BLCA patients [21]. Except for METTL1 and WDR4, other m7G-related genes were also investigated in previous studies [22,23,24,25,26,27,28,29,30,31,32,33,34,35,36,37,38,39,40,41,42]. The stability and translation of CCNB1 are required for specific tRNA m7G methylation, and CCNB1 favors a progressing development in BLCA [24,25]. The interaction of EIF4E3 with m7G cap can recognize the m7G and guanosine [26]. EIF4G1 consists of two complexes, i.e., m7G-cap-binding eIF4F complex and combination with eIF1 [27]. GEMIN5 is capable of binding to m7G cap and is regarded as a m7G cap-binding protein [28]. AP-1 can promote the activity of EIF4E, a m7G cap-binding gene [29]. Critically, AP-1 also serves as a mediator in the progression of BLCA [30]. MYC affects many cellular processes and regulates m7G methyltransferase WDR4 transcription [24]. Additionally, MYC is demonstrated to be a tumor promoter in BLCA [31]. cN-IIIB is a metabolite of mRNA cap, which often expresses a unique reaction to 7-mehtylguanosine monophosphate [32]. GTF2F2 participates in the process of initiating gene transcription and appears in various human tissues and organs [33]. RPS12 improves the translation efficiency of transcripts through hypoxic monosomes [34]. It was found that down-regulation of CYFIP2 was related to a poor prognosis of BLCA patients [35]. Nur77 was previously confirmed to be a promising target for the management of BLCA [36]. Likewise, MTH1 has the potential to be a chemotherapeutic target for advanced BLCA patients and refractory patients receiving cisplatin-based combination chemotherapy [37]. Knockdown of PNP suppresses the cell activity and induces apoptosis in BLCA [38]. TNPO1 regulates the alternative splicing in BLCA [39]. Similar to TNPO1, LENG8 is an alternative splicing-associated prognostic biomarker in kidney renal clear cell carcinoma patients [40]. Grisanzio et al. corroborated that NUDT11 was a tumor initiator in prostate cancer, which impacted tumor-related cellular phenotypes [41]. Methylation of PRKCB can independently predict the recurrence of prostate cancer [42].

More recently, with the increased understanding of m7G’s role in cancer, a number of studies are focusing on its role in shaping tumor microenvironment. Nowadays, two major forms of m7G (m7G-related subtypes and m7G-related signatures) are used to investigate the association with tumor immune infiltrating cells. On the one hand, some studies found that the m7G-related regulators could impact the distribution of some immune cells among different m7G-associated subtypes. Wu et al. noted that a high proportion of immune cells was enriched in C3 m7G-related subtype in glioblastoma [43]. Dong et al. observed that clear cell renal cell carcinoma patients in the CS2 m7G-related group expressed a lower abundance of immune cells, such as naïve B cells, CD8^+^ T cells, follicular helper T cells and M0, M1 and M2 macrophages [44]. On the other hand, m7G-related mRNA, lncRNA and microRNA were utilized in some previous studies to construct corresponding m7G-related signatures associated with tumor immune microenvironment. For example, prostate cancer patients in the high-risk group tended to have less activated dendritic cells and neutrophils compared to those in the low-risk group [45]. More activated dendritic cells, CD8^+^ T cells, neutrophils and Tregs were noticed in colon cancer patients with low m7G-related lncRNA risk scores [46]. The infiltrating levels of activated dendritic cells, follicular helper T cells and Tregs were negatively correlated with m7G-related microRNA risk scores in breast cancer [47].

As for the relationship between m7G methylation and response to some chemotherapeutic drugs, it was reported that decreased resistance to cisplatin and docetaxel was associated with stable knockdown of METTL1 in cervical cancer [48]. The relationship between m7G and immune response has also been unearthed in some studies. An immunosuppressive microenvironment was formed with the participation of METTL1 and WDR4, providing more guiding approaches for future immunotherapy [17,49].

In this study, we integrated three cohorts and used a consensus unsupervised clustering method to classify the BLCA patients into different clusters based on 18 m7G-related genes. To reveal the potential differences among these m7G-related clusters, we identified some candidate proteins via various approaches and developed a novel scoring system based on the GSVA algorithm. Thus, we decided to investigate if this novel constructed m7G-related signature could be utilized for clinically significant patient risk stratification and also for selection, in the preclinical setting, of potentially responsive patients in order to design targeted therapeutic approaches.

## 2. Materials and Methods

### 2.1. Data Collection and Procession

The RNA-seq data of BLCA patients with corresponding clinical information in GSE13507 and TCGA-BLCA datasets were separately downloaded from GEO (http://www.ncbi.nlm.nih.gov/geo/ (accessed on 1 March 2022)) and TCGA (https://xenabrowser.net/ (accessed on 1 March 2022)) databases. Additionally, transcriptional data and corresponding clinical features of metastatic BLCA patients receiving anti-PD-L1 immunotherapy were obtained using the “IMvigor210CoreBiologies” package [50]. Gene expression profiles were measured by the transcript per million estimation and log_2_-based transformation. Subsequently, mRNA expressions of BLCA patients in these three cohorts were merged and batch-corrected via the “sva” package [51]. The detailed clinical information of BLCA patients is shown in Table 1. Eighteen m7G-related genes (CCNB1, CYFIP2, EIF4E3, EIF4G1, GEMIN5, GTF2F2, JUN, LENG8, MYC, NR4A1, NT5C3B, NUDT1, NUDT11, PNP, PRKCB, RPS12, TNPO1 and WDR4) were collected based on the keywords “7-methylguanosine”, “N7-methylguanosine” and “m7G” from GSEA (https://www.gsea-msigdb.org/, accessed on 5 March 2022) and Genecards (https://www.genecards.org/, accessed on 5 March 2022) databases. Detailed information on the 18 m7G-related genes is shown in Appendix A. The overview design of this study is displayed in Figure 1.

### 2.2. Consensus Unsupervised Clustering

We used the “ConsensusClusterPlus” package to perform consensus unsupervised clustering analysis based on expressions of 18 m7G-related genes identifying different m7G-related clusters [52]. We applied the “pam” algorithm with “euclidean” as a measure of distance accompanied with 80% item resampling and 1000 repetitions. The optimal k was determined according to the proportion of ambiguous clustering (PAC) and appeared when the PAC attached the lowest value [53]. The “limma” package was adopted to determine genes differentially expressed among different m7G-related clusters with a |log_2_FC| > 1 and *p*-adjusted < 0.05 [54].

### 2.3. Quantification of Tumor Immune Microenvironment

Twenty-two immune cells characterizing the tumor immune microenvironment were assessed through the “CIBERSORT” algorithm [55]. The fraction of 22 immune cells was calculated based on normalized gene expression profiles and leucocyte signature matrix 22 followed by running with 1000 permutations. Patients with *p* < 0.05 were considered for further analysis. In addition, we scored the tumor microenvironment through “xCell” and “ESTIMATE” algorithms [56,57]. The estimate scores were comprehensively used to evaluate the tumor microenvironment, in which the immune scores and stromal scores respectively reflect tumor immune components and stromal components.

### 2.4. Functional Enrichment Analysis

We conducted “GO” enrichment and “KEGG” pathway analyses to disclose the biological function of certain genes differentially expressed among different m7G-related clusters. The “clusterProfiler” package with a set of “BH” method and FDR < 0.05 was required for this process [58]. The “GO” enrichment functions included cellular components, biological property and molecular function.

### 2.5. Development and Validation of a m7G-Related Scoring System

Based on these genes differentially expressed among different m7G-related clusters, we performed the second clustering analysis. Among these differentially expressed genes (DEGs) relevant to m7G-related clusters, we defined the genes positively correlated with the m7G-related clusters signature as gene signature A, and the genes negatively correlated with the m7G-related clusters signature were defined as gene signature B. Univariate Cox analysis was used to determine the prognostic value of these DEGs with the “survival” package. The prognostic genes were screened with the criteria of *p* < 0.05. Then, LASSO was used to decrease the multicollinearity to acquire some important genes using “glmnet”. Finally, the “GSVA” method was employed to calculate the feature scores of these important gene sets in signatures A and B [59]. A novel m7G-related scoring system was developed according to the following formula: GSVA_A_–GSVA_B_. The optimal cutoff value originated from the “surv_cutpoint” function in the “survminer” package. Next, BLCA patients were divided into high- and low-score groups based on the optimal cutoff value. TCGA-BLCA, GSE13507 and IMvigor210 cohorts were used to evaluate the discriminated power of this established m7G-related signature. Apart from this, survival status maps and risk curves were simultaneously plotted in order to delve into the relationship between this m7G-related signature and cancer development.

### 2.6. Comparison with Clinical Features

All clinical features containing age, gender, N stage, T stage, stage and grade were incorporated with the established m7G-related score. We investigated whether this constructed m7G-related signature was an independent prognostic indicator in the process of comparison with other clinical features through presenting univariate and multivariate forest plots.

### 2.7. Prediction of Immunotherapy

The expressions of three represented immune checkpoints (ICs) were compared between high- and low-score groups. Beyond that, we utilized the Tumor Immune Dysfunction and Exclusion (TIDE) algorithm to explore the role of this m7G-related score in predicting the BLCA patients’ response to immunotherapy. The TIDE score comprehensively reflected the tumor immune dysfunction and exclusion [60]. The higher level of TIDE, dysfunction and exclusion scores indicated a worse response to immunotherapy. Finally, a real immunotherapy cohort of BLCA patients was used to validate this applicable value of our signature.

### 2.8. Drug Sensitivity Analysis

To fully exert the potential of this signature in targeted chemotherapy, we used “pRRophetic” to calculate the IC50 value of three classic chemotherapeutic drugs effectively used in the treatment of BLCA [61]. We compared the IC50 values of cisplatin, docetaxel and paclitaxel between high- and low-score groups, so as to test the patients’ sensitivity to each drug.

### 2.9. Statistical Analysis

All analytical processes were performed based on R4.0.3. The Kaplan–Meier plot survival with the log-rank test was utilized to compare the overall survival (OS) of different groups. The Kruskal–Wallis and Wilcoxon tests were applied for the significance test in comparison of different groups. Spearman correlation analysis was used for revealing the relationship between m7G-related scores and immune, stromal and estimate scores. *p* < 0.05 was considered a statistical difference.

## 3. Results

### 3.1. Identification of m7G-Related Clusters

The PCA analysis showed that there was an obviously decreased batch effect after correction (Figure 2A,B). We classified 760 BLCA patients into three clusters (cluster 1: 205 samples, cluster 2: 218 samples, cluster 3: 337 samples) when k = 3 had the lowest PAC value (Appendix A and Appendix A). Kaplan–Meier survival analysis indicated that patients in cluster 2 had a worse prognosis than those in both cluster 1 (*p* = 0.0053) and cluster 3 (*p* = 1 × 10^−4^). There was no statistical OS difference between patients in clusters 1 and 3 (*p* = 0.45) (Figure 2C). The expression levels of 18 m7G-related genes were displayed via a comprehensive heatmap among the three clusters (Figure 2D). To explore the tumor microenvironment of the three clusters, we performed the “CIBERSORT” algorithm. Differentially enriched immune cells showed that the patients in cluster 3 had a higher fraction of naïve B cells, plasma cells, CD8^+^ T cells, regulatory T cells (Tregs) and monocytes (Figure 2E). On the other hand, greater proportions of activated memory CD4^+^ T cells, resting NK cells, M0 and M1 macrophages and activated mast cells were found in cluster 2 (Figure 2E). We conducted differentially expressed analyses to reveal the differences among the three clusters. There were 418 DEGs between cluster 1 and 2, 643 DEGs between cluster 1 and 3 and 526 DEGs between cluster 2 and 3 (Appendix A). After removing duplicated genes, we obtained 1108 DEGs in total. The participated functions of these DEGs were associated with immune-related activities, such as regulation of immune effector process, regulation of innate immune response, immune receptor activity, cytokine receptor activity, Th1 and Th2 cell differentiation and cytokine–cytokine receptor interaction (Figure 3A,B). To our surprise, these genes were related to the development of BLCA, suggesting their potential to target BLCA (Figure 3B). Next, we utilized consensus unsupervised clustering based on the profile of the 1108 genes, so that 760 BLCA patients could clearly be divided into two clusters, cluster 1 (329 samples) and cluster 2 (431 samples) (Appendix A and Appendix A). Cluster 2 was linked to a better OS (Figure 3C). Similarly, different expression levels of 18 m7G-related genes between these two groups were displayed via a heatmap (Figure 3D). Subsequently, we detected a higher infiltration of naïve B cells, plasma cells, CD8^+^ T cells, follicular helper T cells and Tregs in cluster 2, and more activated memory CD4^+^ T cells, activated NK cells, M0, M1 and M2 macrophages and neutrophils were gathered in cluster 1 (Figure 3E).

### 3.2. Construction and Validation of a Novel m7G-Related Scoring System

In 1108 DEGs relevant to m7G-related clusters, 257 genes positively correlated with the cluster signature were termed as gene signature A. Additionally, 851 genes negatively correlated with the cluster signature were defined as gene signature B. Univariate Cox analysis screened out 114 prognostic genes in signature A and 340 prognostic genes in signature B (Appendix A). To reduce the multicollinearity and dimension, we applied the LASSO algorithm for the selection of key genes. We further acquired 11 genes without zero coefficients in signature A and 29 genes without zero coefficients in signature B (Appendix A and Appendix A). Finally, we computed the m7G-related scores of 760 BLCA patients according to the GSVA formula above. The distinct OS differences were observed in the whole test cohort and three independent test cohorts (Figure 4A,D,G,J). Patients with higher scores tended to be associated with a favorable prognosis. Meanwhile, the risk curves and survival status maps emphasized that this m7G-related signature had strong discriminatory power (Figure 4B,C,E,F,H,I,K,L).

### 3.3. The m7G-Related Signature Was an Independent Predictor

The results of univariate Cox regression indicated that age, N stage, T stage, stage and m7G-related signature were associated with OS in the TCGA-BLCA dataset (Figure 5A). Age, grade and m7G-related signature were related to OS in GSE13507 (Figure 5C), whereas gender and m7G-related signature were relevant to OS in the IMvigor210 cohort (Figure 5E). Following the analysis by multivariate Cox regression, we ascertained that age and m7G-related signature remained prognostic indicators for BLCA patients in TCGA-BLCA and GSE13507 (Figure 5B,D). Gender and m7G-related signature were protective factors for the prognosis of BLCA patients in the IMvigor210 cohort (Figure 5F). Therefore, these findings demonstrate that this m7G-related signature could predict the OS of BLCA patients with the independence of other clinical features.

### 3.4. The Correlation with Tumor Microenvironment

We further explored the role of this m7G-related signature in the tumor microenvironment. We found a higher abundance of naïve B cells, plasma cells, CD8^+^ T cells, follicular helper T cells and Tregs but a lower fraction of M0 and M2 macrophages, activated mast cells and neutrophils in patients with a high score compared to those in the low-score group (Figure 6A). The ESTIMATE results showed that m7G-related scores were negatively associated with the immune, stromal and estimate scores (Figure 6B–D). The xCell algorithm was employed to ensure the stability of these results. As a result, the immune, stromal and estimate scores were also negatively correlated with the m7G-related scores (Figure 6E–G).

### 3.5. Potential in Prediction of Immunotherapy and Targeted Chemotherapeutic Drugs

To reasonably choose which patients were more suitable for receiving immunotherapy and targeted chemotherapeutic drugs, we compared some clinical indicators and biomarkers between high- and low-score groups. Firstly, the levels of three common ICs (CD274, CTLA4 and PDCD1), TIDE and T cell exclusion scores were significantly elevated in the low-score group (Figure 7A–F,H,I,K,L). Although no significant correlation between T cell exclusion scores and m7G-related scores was found in GSE13507, an obvious up-regulation of T cell exclusion scores appeared in the low-score group in the TCGA-BLCA dataset (Figure 7G,J). To test the potential of this signature in the prediction of immunotherapy from a real immunotherapy cohort, we chose 195 metastatic BLCA patients receiving anti-PD-L1 immunotherapy. The results showed that CD274, CTLA4 and PDCD1 were the up-regulated ICs in the low-score group (Figure 8A–C). Patients responding to immunotherapy had higher m7G-related scores, and the overall response rate was higher in the high-score group compared to the low-score group (Figure 8D,E). These results suggest that patients in the high-score group had a higher probability of response to immunotherapy. Considering that some patients were resistant to some classic drugs, three classic chemotherapeutic drugs were introduced into this study. Interestingly, we found that all high-score patients in three cohorts had higher IC50 values than those in the low-score group, highlighting that those low-score patients were more sensitive to these classic drugs (Figure 9).

## 4. Discussion

It was demonstrated that m7G impacted the development of tumors and patients’ response to targeted therapies in many studies [13,14,15,16,17,18,19,48]. The high heterogeneity of BLCA was the main cause of different clinical outcomes and sensitivities to clinical treatments. Considering that m7G modification favors the BLCA progression, we developed a novel m7G-related scoring system for risk stratification and for the prediction of personalized therapies.

Subtype-based recognition exerts a vital role in the clinical intervention in BLCA. Although the correlation between m7G modification and BLCA has been disclosed in previous studies, there is limited understanding of the comprehensive effect of m7G-related characterization in BLCA. In this study, we divided 760 BLCA patients from different platforms into three clusters. Patients in cluster 2 displayed the worst OS compared to those in cluster 1 and cluster 3, indicating that these m7G-related genes may affect the prognosis of BLCA. To investigate the differences in tumor immune microenvironment, we compared the infiltrating levels of 22 tumor immune cells among these clusters. We discovered that cluster 3 patients were associated with a longer OS and had a higher proportion of plasma cells, CD8^+^ T cells and Tregs. CD8^+^ T cells were protective indicators in the process of anti-tumor immunity in various tumors, including BLCA [62,63]. Gu et al. found that low-risk BLCA patients exhibited a higher immune infiltration of plasma cells compared to the high-risk group, which was in line with our results [64]. Interestingly, the Tregs showed an inverse effect in BLCA, though they were associated with decreased survival in many cancers, including breast, melanoma and lung [65,66]. The diverse distribution of immune cells indicated that some potential proteins should be identified. Therefore, we extracted these DEGs on the basis of three m7G-related clusters. Surprisingly, we found that these genes were enriched in some immune-related functions, such as regulation of immune effector process, regulation of innate immune response and Th1 and Th2 cell differentiation, suggesting their roles in targeting in immunotherapy. Most importantly, the participatory activities were involved in BLCA, except for those related to immunity, highlighting a correlation with the development of BLCA.

We subsequently classified patients into two clusters on the basis of these DEGs associated with m7G-related clusters. The results showed that cluster 2 patients had a better prognosis and a high infiltration of naïve B cells, plasma cells, CD8^+^ T cells, follicular helper T cells and Tregs. Next, we constructed a m7G-related signature based on some crucial DEGs relevant to m7G-related clusters according to the principle of the GSVA algorithm to biologically elucidate the impact of m7G-related characterization on the prognosis of BLCA. This m7G-related signature could effectively be used for risk stratification and independently predict the prognosis of BLCA by integrating other clinical features in various cohorts. Patients with higher m7G-related scores had a better OS than those with lower m7G-related scores. Similar to the results above, a higher abundance of plasma cells, CD8^+^ T cells, follicular helper T cells but a lower fraction of M0 and M2 macrophages, activated mast cells and neutrophils were gathered in the high-score group compared to the low-score group. In addition, patients with higher immune, stromal and estimate scores presented significantly negative relevance to low m7G-related scores. In previous studies, BLCA patients with m6A cluster 1 who were considered as an immune-excluded phenotype experienced higher immune and estimate scores, showing a worse prognosis [67]. The tumor stromal-infiltrating microenvironment has been considered as a dangerous signal for BLCA patients [68]. According to these results, m7G-related scores were negatively linked to immunosuppressive microenvironment scores, which can explain why patients in the high-score group exhibited an advantage of OS.

Encouraging data from clinical studies indicate that immunotherapeutic strategies could become a major treatment strategy in BLCA. In this respect, the possibility of identifying early on, before starting treatment, which patients will respond to immunotherapy, is a major clinical need. A recent paper showed the possibility of patient-specific immunoprofiling of BLCA cases in order to select which patients will respond to immunotherapy, but no specific biomarkers are currently used yet in clinical practice [69]. Available immune-related algorithms have achieved great success in the prediction of immunotherapy in many tumors [60,70]. Currently, we tried to explore the immunotherapeutic value of this signature with the evidence of the TIDE algorithm, a prevalent and common predictive tool. Before this, we first investigated the distribution of three common ICs between high- and low-score groups. An obviously negative connection between m7G-related scores and the expressions of PD-1, PD-L1 and CTLA-4 was observed, further supporting the appearance of immunosuppressive signatures in the low-score group. High expression of PD-L1 can induce a suppressive immune microenvironment and correlate with a poor prognosis of BLCA patients [71,72]. PD-1 negatively regulates immune responses, resulting in the initiation of anti-tumor T cells [73,74]. A high expression of CTLA4 is more likely to promote immune evasion and suppression in BLCA [75]. Subsequently, we discovered that substantially increased TIDE and T cell exclusion scores were displayed in the low-score group, strongly strengthening a lower immune response of patients with low m7G-related scores. These results from a real immunotherapy cohort showed that patients in the high-score group were more appropriate for immunotherapy, and a higher overall response rate was linked with high m7G-related scores, which was consistent with the findings above.

Nowadays, chemotherapy is also one of the most common treatments for BLCA patients. However, not all patients have a high response to chemotherapeutic drugs due to differences in sensitivity to these chemotherapeutic drugs. Therefore, we chose three classic chemotherapeutic drugs which have helped many BLCA patients protect against the progression of tumors [76,77,78]. We evaluated the relationship between m7G-related scores and IC50 values of these classic drugs. Eventually, we detected that IC50 values of the three classic chemotherapeutic drugs were dramatically elevated in the high-score group, indicating that patients’ higher sensitivity was associated with low m7G-related scores. In the past, guidelines emphasized that cisplatin-based chemotherapy should be recommended for these high-risk BLCA patients [79]. In addition, adjuvant chemotherapy with paclitaxel and docetaxel were excellent candidates for high-risk BLCA patients, leading to delaying the development of cancer [80,81]. These studies can support our conclusion to a large extent.

In this work, we used state-of-the-art bioinformatic tools to reveal the role of the m7G-related signature in the prognosis and personalized treatment of BLCA patients, offering some new data and novel findings in the field of BLCA biomarkers and their clinical application. This signature fully considered the combination of m7G-related genes with m7G-related molecular clusters, which is investigable and innovative. Most importantly, our m7G-related signature could better predict these high-grade and advanced-stage patients, bringing more benefits to these patients at high risk in terms of their prognosis and personalized treatments. However, the heterogeneity of the selected samples from public databases was not significant, suggesting that this m7G-related signature needs to be validated in a real cohort with more heterogeneity in the future. Additionally, further validation by drug sensitivity experiments is still needed, though we investigated the relationship between this m7G-related signature and some targeted chemotherapeutic drugs.

Although the clarification of the role of m7G-related modifications in the development of BLCA is still in a preliminary phase, our results suggest that BLCA patients’ prognosis and response to personalized therapies were potentially linked to the characterization of m7G-related clusters. We think these valuable findings can provide more directions for further studies.

## 5. Conclusions

We propose a novel m7G-related scoring system and genetic signature that seem to be able to reliably stratify the risk profile of BLCA patients, with prognostic significance. Moreover, our signature and scoring system appear to be able to select patients who will show a better response to immunotherapy or resistance to chemotherapy. These results could significantly improve the possibility of designing patient-oriented treatment strategies in the future.

## Figures and Tables

**Figure 1 cancers-14-05346-f001:**
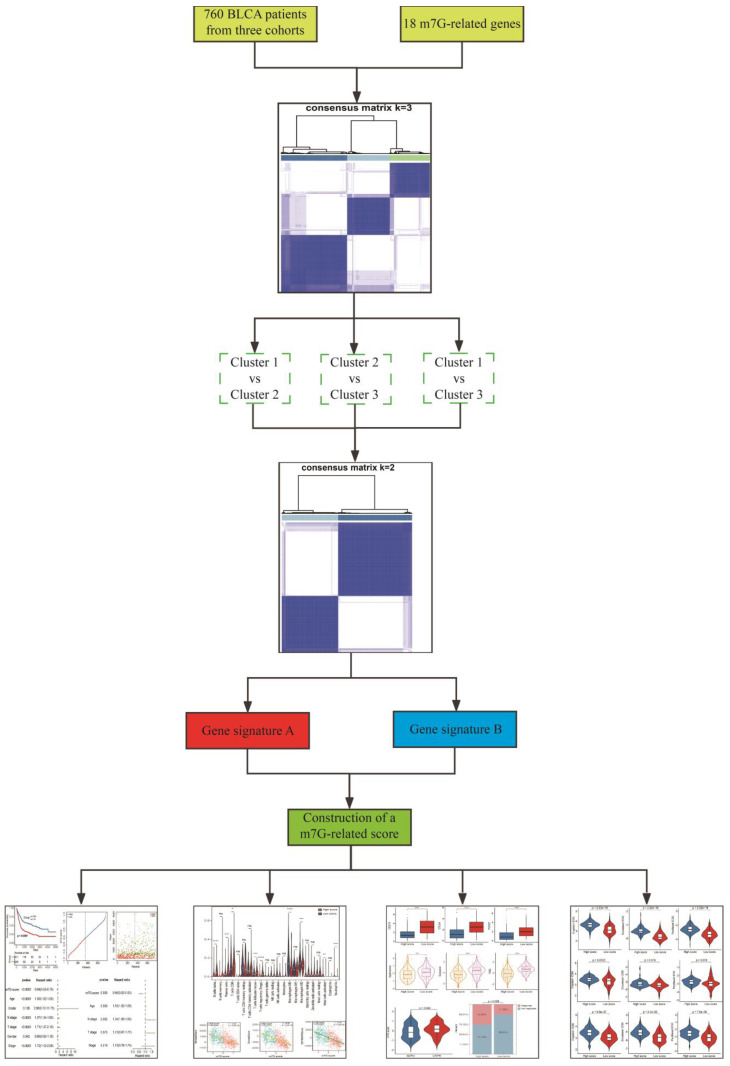
The overview design in this study.

**Figure 2 cancers-14-05346-f002:**
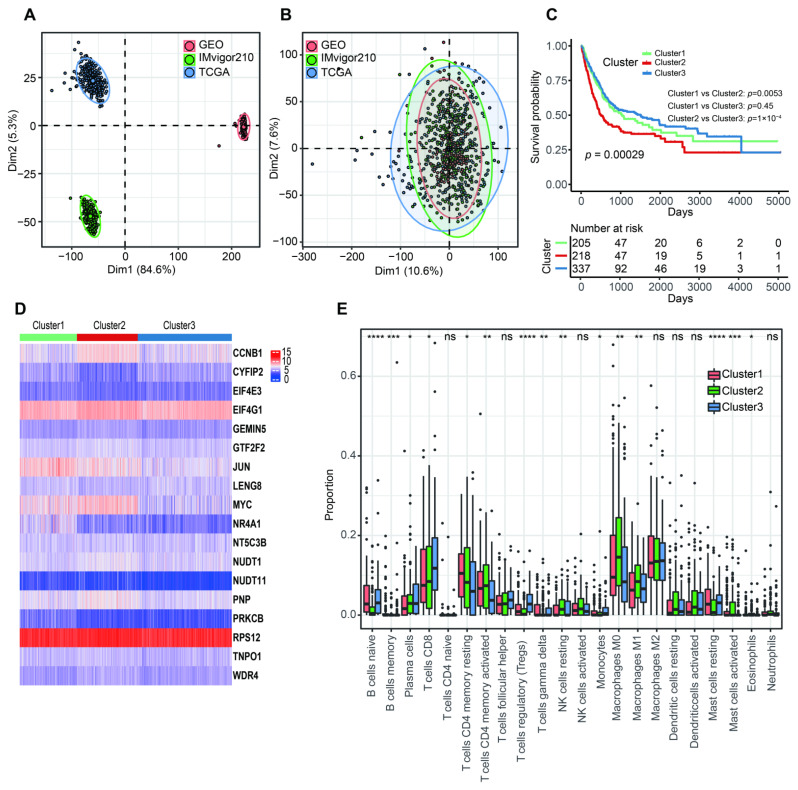
The characteristics of three m7G-related clusters. (**A**) Principal component analysis for the common gene profiles before combination of TCGA-BLCA, GSE13507 and IMvigor210 cohorts. (**B**) Principal component analysis for the common gene profiles after combination of TCGA-BLCA, GSE13507 and IMvigor210 cohorts. (**C**) The Kaplan–Meier survival analysis of OS among three m7G-related clusters. (**D**) The heatmap for distribution of 18 m7G-related genes among three m7G-related clusters. (**E**) The abundance of 22 tumor-infiltrating immune cells in three m7G-related clusters. Data in (**E**) were analyzed by Kruskal–Wallis test; ns, no significance; * *p* < 0.05, ** *p* < 0.01, *** *p* < 0.001 and **** *p* < 0.0001.

**Figure 3 cancers-14-05346-f003:**
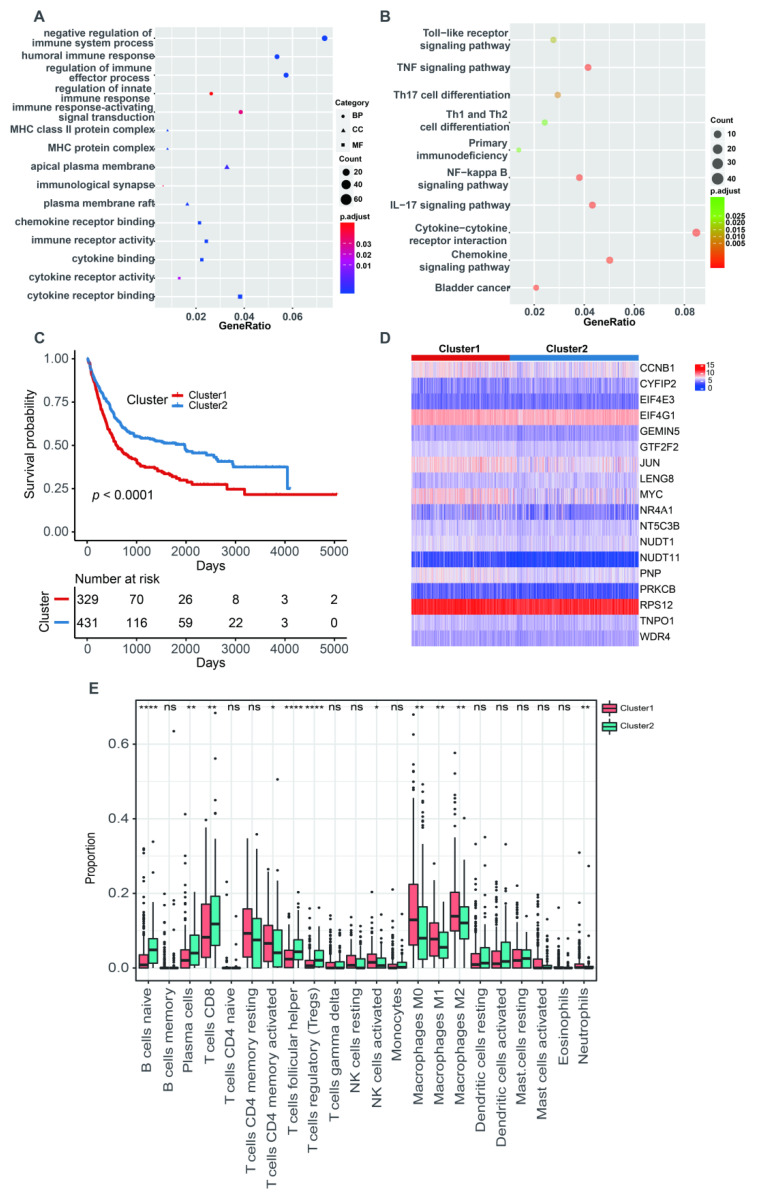
The characteristics of two m7G-related clusters. (**A**) The GO analysis of 1108 DEGs. (**B**) The KEGG analysis of 1108 DEGs. (**C**) The Kaplan–Meier survival analysis of OS between two m7G-related clusters. (**D**) The heatmap for different distribution of 18 m7G-related genes between two m7G-related clusters. (**E**) The abundance of 22 tumor-infiltrating immune cells between two m7G-related clusters. Data in (**E**) were analyzed by Wilcoxon test; ns, no significance; * *p* < 0.05, ** *p* < 0.01, *** *p* < 0.001 and **** *p* < 0.0001.

**Figure 4 cancers-14-05346-f004:**
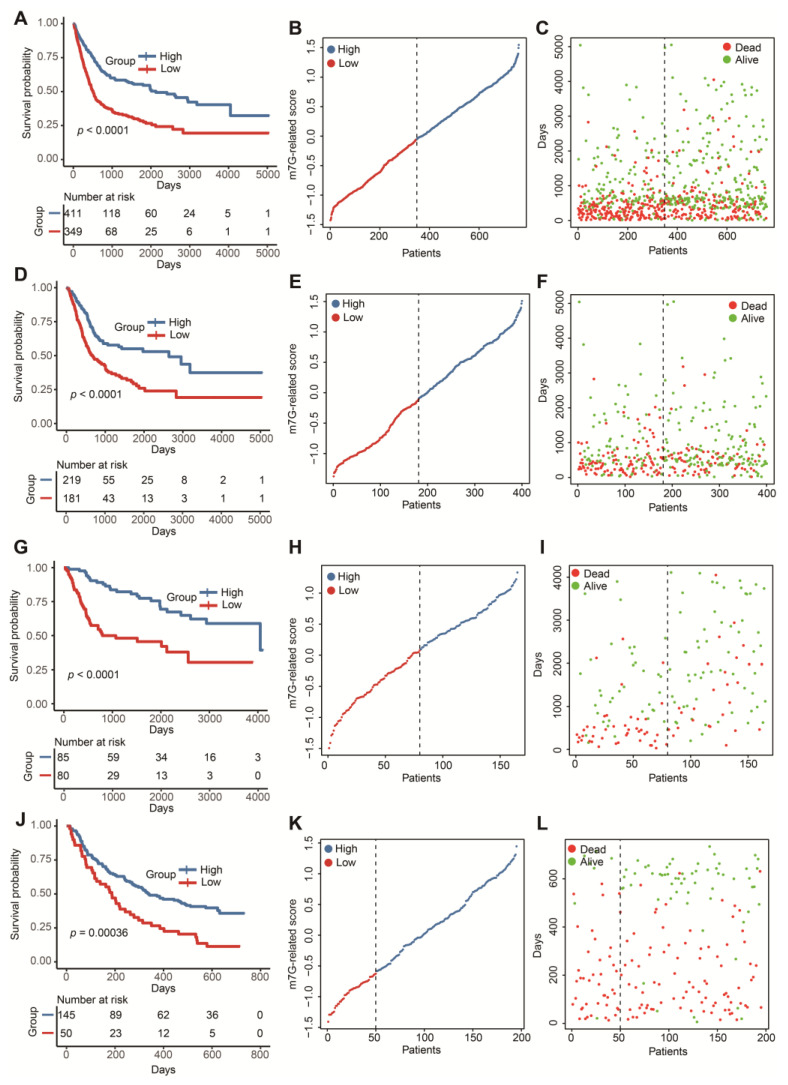
Construction and validation of a m7G-related scoring system. (**A**) The Kaplan–Meier survival analysis of OS between high- and low-score groups in the whole cohort. (**B**) The distribution of m7G-related scores in the whole cohort. (**C**) The relationship between m7G-related signature and survival status in the whole cohort. (**D**) The Kaplan–Meier survival analysis of OS between high- and low-score groups in the TCGA-BLCA dataset. (**E**) The distribution of m7G-related scores in the TCGA-BLCA dataset. (**F**) The relationship between m7G-related signature and survival status in the TCGA-BLCA dataset. (**G**) The Kaplan–Meier survival analysis of OS between high- and low-score groups in the GSE13507 dataset. (**H**) The distribution of m7G-related scores in the GSE13507 dataset. (**I**) The relationship between m7G-related signature and survival status in the GSE13507 dataset. (**J**) The Kaplan–Meier survival analysis of OS between high- and low-score groups in the IMvigor210 cohort. (**K**) The distribution of m7G-related scores in the IMvigor210 cohort. (**L**) The relationship between m7G-related signature and survival status in the IMvigor210 cohort.

**Figure 5 cancers-14-05346-f005:**
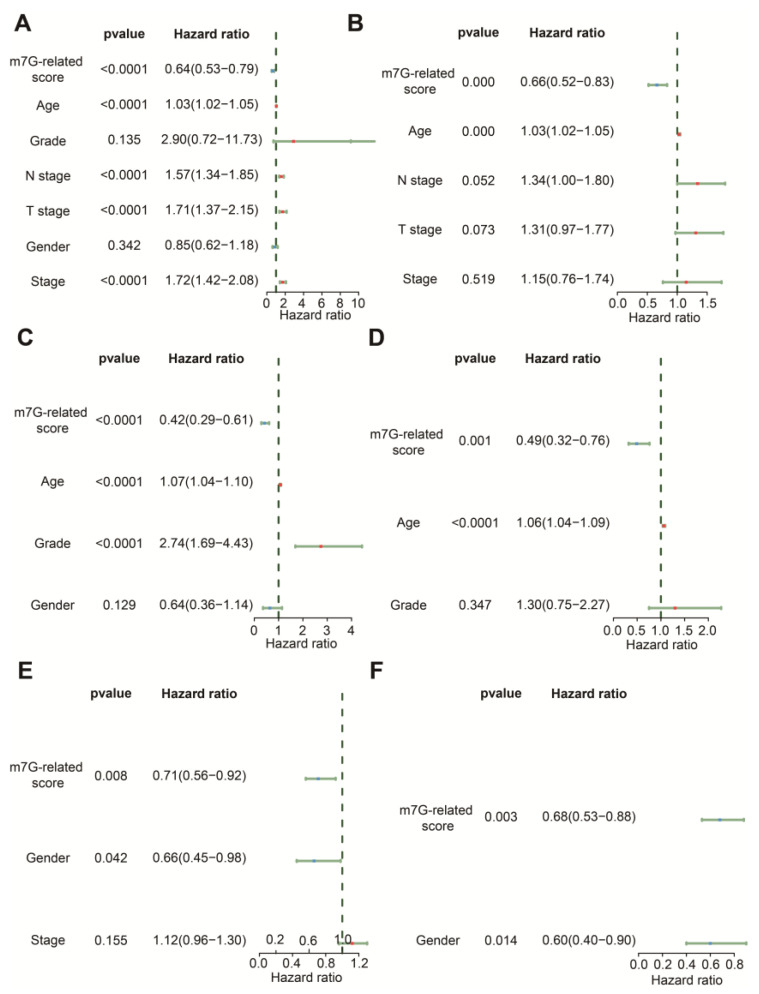
Comparison of other clinical features in prediction of OS. (**A**) The forest plot of univariate Cox regression analysis in the TCGA-BLCA dataset. (**B**) The forest plot of multivariate Cox regression analysis in the TCGA-BLCA dataset. (**C**) The forest plot of univariate Cox regression analysis in the GSE13507 dataset. (**D**) The forest plot of multivariate Cox regression analysis in the GSE13507 dataset. (**E**) The forest plot of univariate Cox regression analysis in the IMvigor210 cohort. (**F**) The forest plot of multivariate Cox regression analysis in the IMvigor210 cohort.

**Figure 6 cancers-14-05346-f006:**
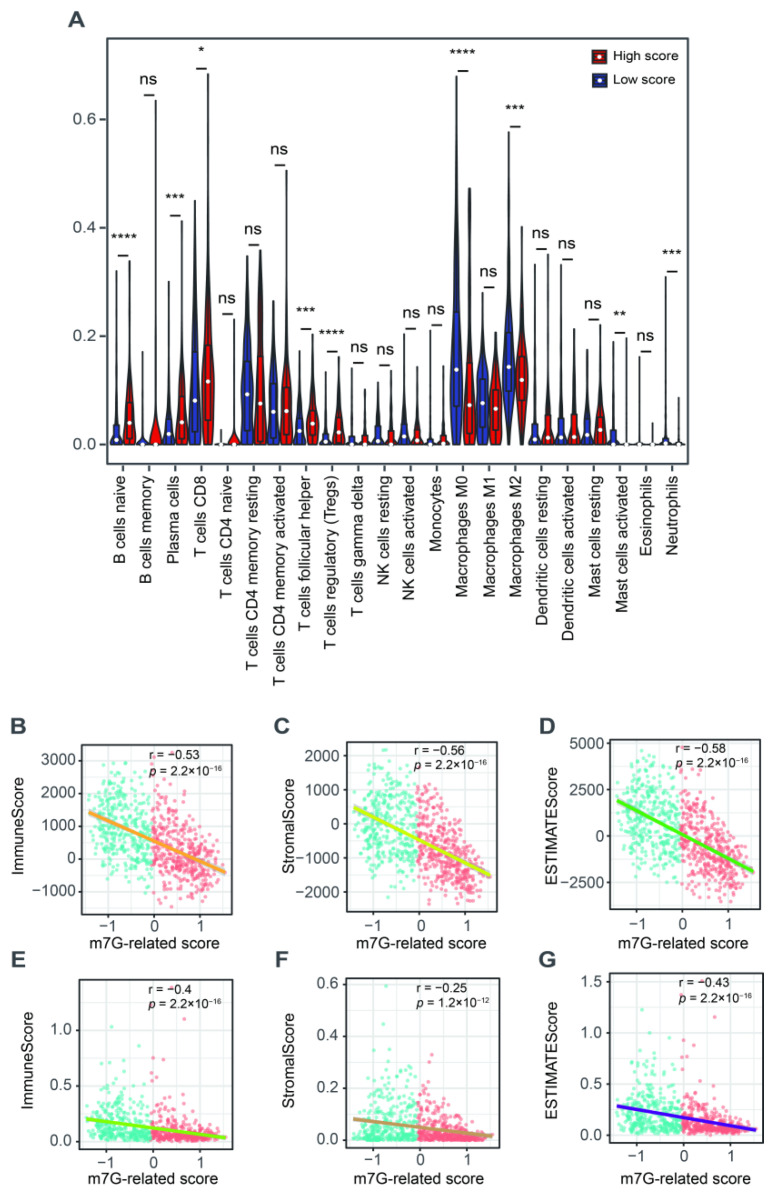
The relationship between m7G-related signature and tumor microenvironment. (**A**) The abundance of 22 tumor-infiltrating immune cells between high- and low-score groups. (**B**) The correlation analysis between immune scores and m7G-related scores via “ESTIMATE” algorithm. (**C**) The correlation analysis between stromal scores and m7G-related scores via “ESTIMATE” algorithm. (**D**) The correlation analysis between estimate scores and m7G-related scores via “ESTIMATE” algorithm. (**E**) The correlation analysis between immune scores and m7G-related scores via “xCell” algorithm. (**F**) The correlation analysis between stromal scores and m7G-related scores via “xCell” algorithm. (**G**) The correlation analysis between estimate scores and m7G-related scores via “xCell” algorithm. Data in (**A**) were analyzed by Wilcoxon test; ns, no significance; * *p* < 0.05, ** *p* < 0.01, *** *p* < 0.001 and **** *p* < 0.0001.

**Figure 7 cancers-14-05346-f007:**
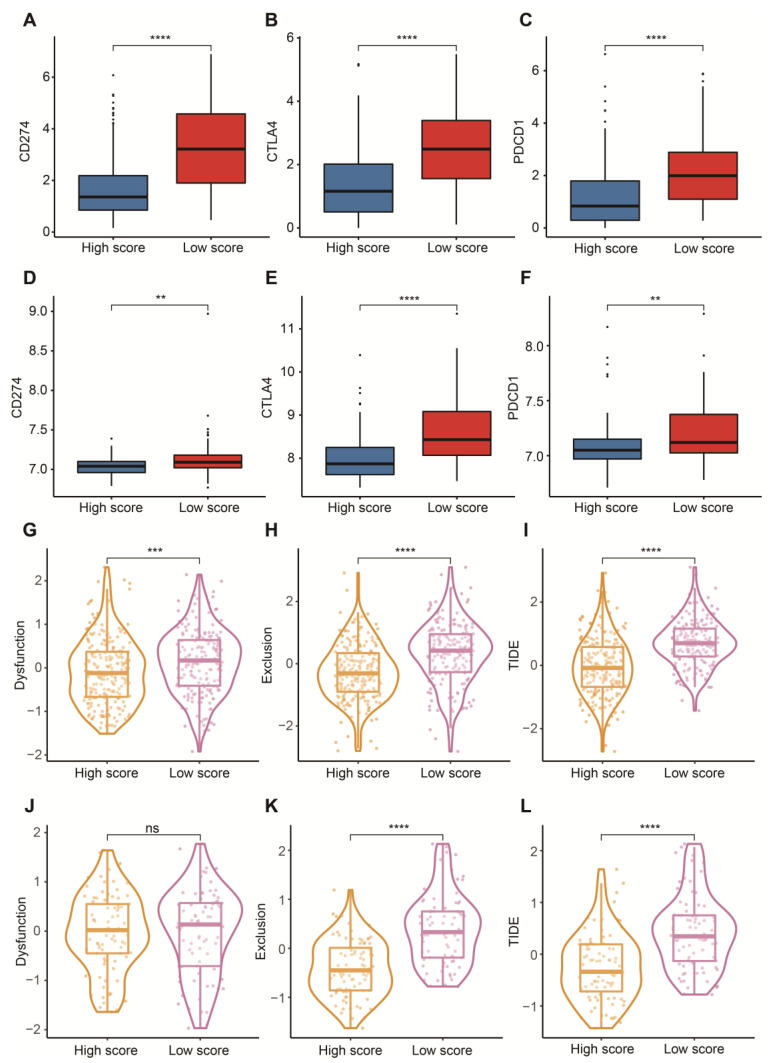
Prediction of immunotherapy. (**A**) Box plots of CD274 between high- and low-score groups in the TCGA-BLCA dataset. (**B**) Box plots of CTLA4 between high- and low-score groups in the TCGA-BLCA dataset. (**C**) Box plots of PDCD1 between high- and low-score groups in the TCGA-BLCA dataset. (**D**) Box plots of CD274 between high- and low-score groups in the GSE13507 dataset. (**E**) Box plots of CTLA4 between high- and low-score groups in the GSE13507 dataset. (**F**) Box plots of PDCD1 between high- and low-score groups in the GSE13507 dataset. (**G**) Distribution of dysfunction scores between high- and low-score groups in the TCGA-BLCA dataset. (**H**) Distribution of exclusion scores between high- and low-score groups in the TCGA-BLCA dataset. (**I**) Distribution of TIDE scores between high- and low-score groups in the TCGA-BLCA dataset. (**J**) Distribution of dysfunction scores between high- and low-score groups in the GSE13507 dataset. (**K**) Distribution of exclusion scores between high- and low-score groups in the GSE13507 dataset. (**L**) Distribution of TIDE scores between high- and low-score groups in the GSE13507 dataset. Data in A–L were analyzed by Wilcoxon test; ns, no significance; ** *p* < 0.01, *** *p* < 0.001 and **** *p* < 0.0001.

**Figure 8 cancers-14-05346-f008:**
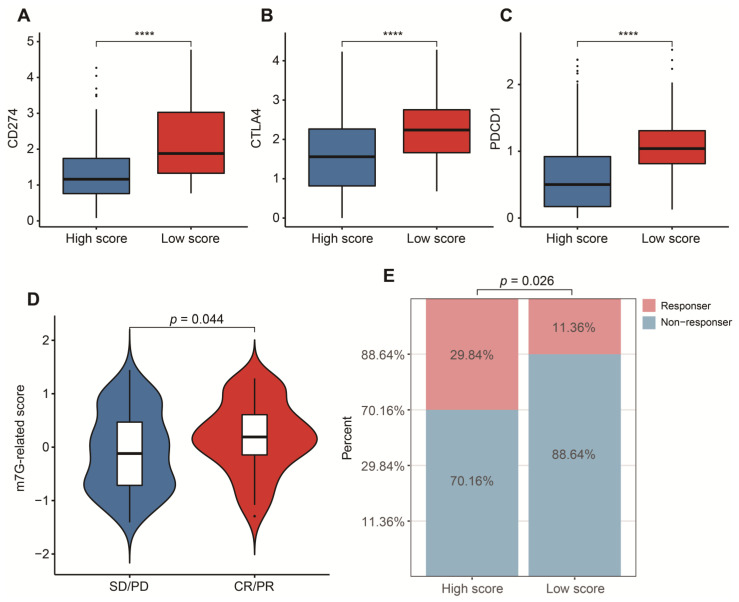
Validation in a real immunotherapy cohort. (**A**) Box plots of CD274 between high- and low-score groups. (**B**) Box plots of CTLA4 between high- and low-score groups. (**C**) Box plots of PDCD1 between high- and low-score groups. (**D**) Distribution of m7G-related scores between responders and non-responders. (**E**) Comparison of overall response rate between high- and low-score groups. Data in (**A**–**C**) were analyzed by Wilcoxon test; **** *p* < 0.0001.

**Figure 9 cancers-14-05346-f009:**
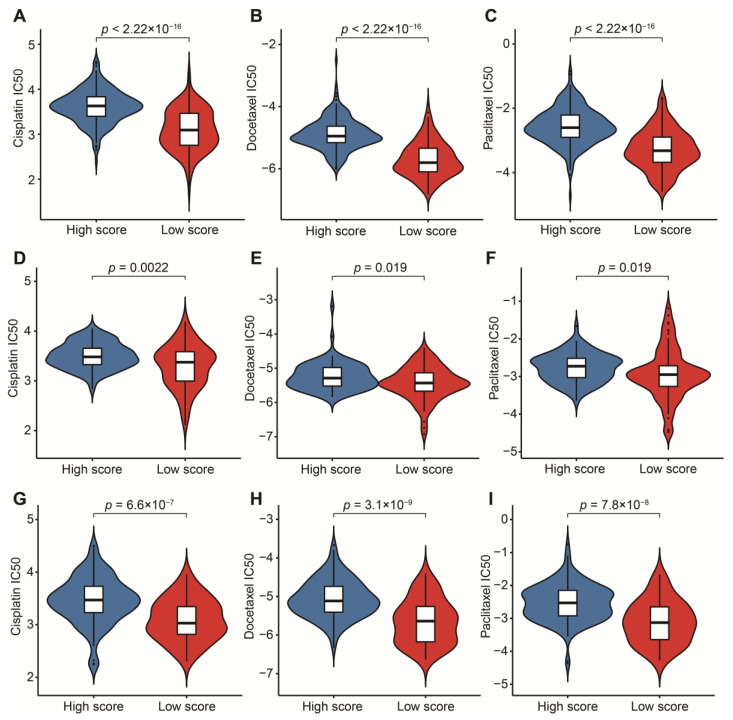
Prediction of sensitivity to three chemotherapeutic drugs. (**A**) Comparison of cisplatin IC50 between high- and low-score groups in the TCGA-BLCA dataset. (**B**) Comparison of docetaxel IC50 between high- and low-score groups in the TCGA-BLCA dataset. (**C**) Comparison of paclitaxel IC50 between high- and low-score groups in the TCGA-BLCA dataset. (**D**) Comparison of cisplatin IC50 between high- and low-score groups in the GSE13507 dataset. (**E**) Comparison of docetaxel IC50 between high- and low-score groups in the GSE13507 dataset. (**F**) Comparison of paclitaxel IC50 between high- and low-score groups in the GSE13507 dataset. (**G**) Comparison of cisplatin IC50 between high- and low-score groups in the IMvigor210 cohort. (**H**) Comparison of docetaxel IC50 between high- and low-score groups in the IMvigor210 cohort. (**I**) Comparison of paclitaxel IC50 between high- and low-score groups in the IMvigor210 cohort.

**Table 1 cancers-14-05346-t001:** The clinical characteristics of 760 BLCA patients enrolled in this study.

Variables	TCGA (n = 400)	GSE13507 (n = 165)	IMvigor210 (n = 195)
**Age**			
≤65	159	74	-
>65	241	91	-
**Gender**			
Female	104	30	42
Male	296	135	153
**Grade**			
Low	20	105	-
High	377	60	-
Unknown	3	0	-
**Stage**			
I	2	-	61
II	128	-	53
III	138	-	39
IV	130	-	42
Unknown	2		
**T stage**			
T0	1	-	-
T1	3	-	-
T2	117	-	-
T3	192	-	-
T4	54	-	-
Tx+Unknown	33	-	-
**N stage**			
N0	233	-	-
N1	44	-	-
N2	74	-	-
N3	7	-	-
Nx+Unknown	42	-	-

## Data Availability

The results shown here are based on data generated by TCGA (https://xenabrowser.net/ (accessed on 1 March 2022)), GEO (http://www.ncbi.nlm.nih.gov/geo/ (accessed on 1 March 2022)), IMvigor210 cohort, GSEA (https://www.gsea-msigdb.org/ (accessed on 5 March 2022)), Genecards (https://www.gsea-msigdb.org/ (accessed on 5 March 2022)) and CIBERSORT (https://cibersort.stanford.edu/ (accessed on 8 March 2022)) databases.

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
