# Peer review of "A Novel m7G-Related Genes-Based Signature with Prognostic Value and Predictive Ability to Select Patients Responsive to Personalized Treatment Strategies in Bladder Cancer"

_cancers, 2022, doi:10.3390/cancers14215346_

Round 1

Reviewer 1 Report

Dear authors,

7-Methylguanosine (m7G) epigenetic modification is widely discussed as a prominent role in cancer as well as the tumor immune microenvironment.

For example, in RCC (Dong, K., Gu, D., Shi, J., Bao, Y., Fu, Z., Fang, Y., Qu, L., Zhu, W., Jiang, A. and Wang, L., 2022. Identification and Verification of m7G Modification Patterns and Characterization of Tumor Microenvironment Infiltration via Multi-Omics Analysis in Clear Cell Renal Cell Carcinoma. Frontiers in Immunology, p.1971.), glioma (Wu X, Li C, Wang Z, Zhang Y, Liu S, Chen S, Chen S, Liu W, Liu X. A bioinformatic analysis study of m7G regulator-mediated methylation modification patterns and tumor microenvironment infiltration in glioblastoma. BMC cancer. 2022 Dec;22(1):1-7.), lung adenocarcinoma (Wang G, Zhao M, Li J, Li G, Zheng F, Xu G, Hong X. m7G-Associated subtypes, tumor microenvironment, and validation of prognostic signature in lung adenocarcinoma. Frontiers in genetics. 2022 Aug 10;13:954840) and others

Nevertheless, such a topic has not been addressed for bladder cancer.

In this article, the authors used m7G-related modifications to develop a novel scoring system. They correlated the m7G modification to patient prognosis, overall survival, infiltrating immune cells, some immunotherapeutic parameters (e.g., PD-1, PD-L1, and CTLA-4), and chemotherapeutic drugs IC50

The results are quite acceptable.

In the introduction, the authors mentioned only METTL1 and WDR4 as two types of genes involved in the methylation process, however, nothing was mentioned about other gene series involved in such a process as well as the 22 types of immune cells.

It is better to add a brief background regarding the genes involved in m7G modification and types of the immune cells used for this study.   

Author Response

Dear reviewer,

Thank you for reviewing our manuscript titled “A novel signature to predict prognosis and tumor microenvironment based on m7G-related genes for bladder cancer”. We have carefully studied the comments and suggestions, and then revised the manuscript accordingly. The changed and added texts in the manuscript are shown in yellow. Also, please note that some tables have been changed slightly, which are different from that in the original one, because we added the one new supplementary table. We hope that the revision could be acceptable, and that our responses adequately address the comments. Should you have any questions, please contact us without hesitation.

7-Methylguanosine (m7G) epigenetic modification is widely discussed as a prominent role in cancer as well as the tumor immune microenvironment.

For example, in RCC (Dong, K., Gu, D., Shi, J., Bao, Y., Fu, Z., Fang, Y., Qu, L., Zhu, W., Jiang, A. and Wang, L., 2022. Identification and Verification of m7G Modification Patterns and Characterization of Tumor Microenvironment Infiltration via Multi-Omics Analysis in Clear Cell Renal Cell Carcinoma. Frontiers in Immunology, p.1971.), glioma (Wu X, Li C, Wang Z, Zhang Y, Liu S, Chen S, Chen S, Liu W, Liu X. A bioinformatic analysis study of m7G regulator-mediated methylation modification patterns and tumor microenvironment infiltration in glioblastoma. BMC cancer. 2022 Dec;22(1):1-7.), lung adenocarcinoma (Wang G, Zhao M, Li J, Li G, Zheng F, Xu G, Hong X. m7G-Associated subtypes, tumor microenvironment, and validation of prognostic signature in lung adenocarcinoma. Frontiers in genetics. 2022 Aug 10;13:954840) and others

Nevertheless, such a topic has not been addressed for bladder cancer.

In this article, the authors used m7G-related modifications to develop a novel scoring system. They correlated the m7G modification to patient prognosis, overall survival, infiltrating immune cells, some immunotherapeutic parameters (e.g., PD-1, PD-L1, and CTLA-4), and chemotherapeutic drugs IC50

The results are quite acceptable.

In the introduction, the authors mentioned only METTL1 and WDR4 as two types of genes involved in the methylation process, however, nothing was mentioned about other gene series involved in such a process as well as the 22 types of immune cells.

It is better to add a brief background regarding the genes involved in m7G modification and types of the immune cells used for this study.   

Response: Thanks for your suggestion. We have added a background regarding the genes involved in m7G modification and types of the immune cells used for this study into the revised manuscript (Line 71-113, Page 2-3).

Reviewer 2 Report

Major concerns

1. 

The Title is not in agreement with their actual studies. “The landscape of m7G patterns predicts prognosis …” implicates that this study would investigate the m7G patterns at a global scale and how the landscape of this modification predicts prognosis. However, there was barely anything about m7G modification in the study. In fact, the only link to m7G, which is weak and vague, is a list of 18 genes collected from GSEA. The authors appeared to firmly believe that these genes can be used to represent m7G landscape simply because they are annotated m7G-related regulators in GSEA.

If the authors do GO and KEGG analysis of these genes, they will find that these genes are enriched in many different biological processes, such as ErbB, MAPK, and Wnt signaling. They are also enriched in gene expression. In terms of modification, there are 11 out of these 18 genes related to acetylation. Based on the authors’ logic, we can also state that these are acetylation-related subtypes, signatures, and scores.

This represents one major concern of this study. There is no any evidence to support that the patients clustered in the same group are more similar to each other in terms of m7G modification than those in other groups. As a result, all analyses performed, including the terms they defined such as m7G signatures, scores, and subtypes, are misleading and unjustified.

2.

All analyses performed in this study were derived from those 18 m7G-related genes. The study used these genes to cluster the patients. They found that the clusters are correlated with OS, at least that Cluster 2 is different. Then, DEGs were identified based on these clusters. Using these DEGs, another clustering analysis was performed.

Such an experimental design indicates that the DEGs are related to those 18 m7G-related genes and that the two clustering analyses are correlated.

Several questions have not been addressed in the manuscript regarding the relationships between the two clustering analyses.

In both clustering analyses, one cluster was associated with poor OS. The questions are: are these two clusters related, what is the degree of overlap between the clusters, and is the overlap significant. If not significant, how that be explained. If significant, what is the rationale for doing both clustering analyses and what additional insights can be obtained.

Minor concerns

1. Line 200: “the heatmap showed that 18 m7G-related genes were distributed between two groups”.

What does this statement mean? It has no specific information about the distribution, so it can be considered meaningless. Each gene should be present in each group. The difference is their expression levels among the patients in each group. 

2. The Methods lack sufficient details, and it is confusing about how the experiments were performed.

For example, Method 2.5: “Univariate Cox analysis was used to select these prognostic m7G-related genes depended on “survival” package”.

From what those genes were selected? 18 m7G-related genes? Or 1108 DEGs? It was not described in the text either. 

4. Line 222: “257 gene signatures positively correlated with the number of cluster were termed as the m7G-related gene signature A”.

The statement is not clear. What cluster it refers to? There were two clustering analyses at this point. Which one?

What does it mean “positively correlated with the number of cluster”?

What does it mean “the residual genes were inversely defined as the 223 m7G-related gene signature B”?

5. Line 346: “A distinct OS among these subtypes suggested that the prognosis of BLCA was affected by m7G-related molecules”. This statement might not be completely correct if Cluster 1 and Cluster 3 have no significant difference. Visually, the difference between Cluster 1 and Cluster 3 appears to be not significant (Figure 2C). Are they significantly different? Only one p-value was provided. There was no information about p for which comparision.

6. The authors formulated this study around m7G modification. However, m7G modification was poorly introduced. What are the biochemistry and biology of this modification? What are its targets (mRNA, scRNA, lncRNA, tRNA, rRNA …). It also lacks information about how those 18 m7G genes collected from GESA are related to m7G modification.

7. Line 225: Where is Figure S6?

Reviewer 3 Report

Lai et al have analyzed the patterns of m7G in bladder cancer and used clustering methods to classify patients with bladder cancer and identify an m7G-related signature. The authors also suggest using this signature to predict response of patients to varied therapeutic approaches such as immunotherapy and chemotherapy. Overall, this is a strong study and apart from a few grammatical errors and typos, I recommend publication of this study. 

Author Response

Dear reviewer,

Thank you for reviewing our manuscript titled “A novel signature to predict prognosis and tumor microenvironment based on m7G-related genes for bladder cancer”. We have carefully studied the comments and suggestions, and then revised the manuscript accordingly. The changed and added texts in the manuscript are shown in yellow. Also, please note that some tables have been changed slightly, which are different from that in the original one, because we added the one new supplementary table. We hope that the revision could be acceptable, and that our responses adequately address the comments. Should you have any questions, please contact us without hesitation.

Lai et al have analyzed the patterns of m7G in bladder cancer and used clustering methods to classify patients with bladder cancer and identify an m7G-related signature. The authors also suggest using this signature to predict response of patients to varied therapeutic approaches such as immunotherapy and chemotherapy. Overall, this is a strong study and apart from a few grammatical errors and typos, I recommend publication of this study. 

Response: Thanks for your recommendation. These grammatical errors and typos have been revised by a professional in English writing. The corresponding revised parts were shown in revised manuscript (Line 32, Page 1; Line 64; Line 67-68, Page 2; Line 134-135, Page 3; Line 210, Page 6; Line 234; Line 253, Page 7; Line 268, Page 8; Line 366, Page 13; Line 424; Line 450, Page 17).

Reviewer 4 Report

This manuscript is interesting and offers some new data and novel findings in the field of bladder cancer biomarkers and their clinical application. However, it needs a number of changes before being accepted for publication in Cancers. The changes needed are detailed in the text below.

TITLE: The most important finding of this paper, which is also its main strength, is that the novel signature proposed by the Authors is able to select patients, in the preclinical setting for patient-specific therapeutical strategies in bladder cancer. This is a real novel finding. I think therefore that this fact should be clearly stated in the title of the paper, which is at the moment a bit generic and not very attention-catching. A proposed change for the title, which would make it a real hit, is:

A NOVEL m7G-RELATED GENES-BASED SIGNATURE WITH PROGNOSTIC VALUE AND PREDICTIVE ABILITY TO SELECT PATIENTS RESPONSIVE TO PERSONALIZED TREATMENT STRATEGIES IN BLADDER CANCER

ABSTRACT: LINE 37: In summary, our data show that m7G-related characterization of bladder cancer patients can be of value for prognostic stratification and for patients-oriented therapeutic options, designing personalized treatment strategies in the preclinical setting.

INTRODUCTION:

Line 48 Despite these therapies can improve the prognosis of the disease, a number of patients show a low or no response to treatments, partly because of high tumor heterogeneity. With the development of more advanced methodologies and technologies, molecular subtyping of bladder cancer cases has provided some clues for selecting upfront, before starting treatment, which patients will respond to the appropriate therapies(please cite: Roviello, G. et al. Focus on Biochemical and Clinical Predictors of Response to Immune Checkpoint Inhibitors in Metastatic Urothelial Carcinoma: Where Do We Stand? Int. J. Mol. Sci. 2020,21, 7935. https://doi.org/10.3390/ijms21217935).. Recently, novel approaches based on bioinformatics and machine learning have attracted attention on the possibility of predicting prognosis and selecting appropriate therapies. There is still a need, however, to develop novel genetic signatures with clinical value, in order to transfer the high amount of data obtained with the bioinformatic approach to the clinical setting (reviewed with specific clinical focus in Mancini M., et al. Stem cells, Biomarkers and Genetic Profiling: approaching future challenges in Urology. Urologia J 83:4-13;2016; DOI:10.5301/uro.5000165)

Line 56: not “which the seventh N position” but “where the seventh N position”

Line 57: do not use “is added”, but “is complexed with”

Line 94: More recently, with the increased understanding of m7G role in cancer, a number of studies are focusing on its role in shaping tumor microenvironment.

Line 117-118: erase, unnecessary repetition (from “Thus…..until the end of the sentence).

Line 123: Re-phrase. For example: Thus, we decided to investigate if this novel constructed m7G-related signature could be utilized for clinically significant patients risk stratification and also for selection, in the preclinical setting, of potentially responding patients in order to design targeted therapeutic approaches.

Table 1: I noticed that your group of 760 BLCA patients is not heterogeneous, being the patients mostly “high grade” and in a high stage of disease (mostly stage 3 and 4 cases). I think one of the reasons of the good correlation you are showing between the m7G signature and prognosis and therapy response is that the sample is not very heterogenous. You could have had less significant results with a more heterogenous group, including more low grade, low stage patients (which are, in the clinical real world, the majority of BLCA patients). This fact could be a limitation, but it is also a strength of this study, because the patients with high grade and high stage disease are the patients with the worst prognosis and that more often require additional therapies after or before surgery. You should discuss this point at the end of the discussion section, talking about strengths and limitations of your paper (a paragraph that is missing, but need to be added)

DISCUSSION:

Line 451: Erase “with a breakthrough in immunotherapy “(unnecessary). Start as: Encouraging data from clinical studies indicated that immunotherapeutic strategies could become a major treatment strategy in BLCA. In this respect, the possibility of identifying early on, before starting treatment, which patients will respond to immunotherapy, is a major clinical need. A recent paper showed the possibility of patient-specific immunoprofiling of bladder cancer cases in order to select which patients will respond to immunotherapy, but no specific biomarkers are currently used yet in clinical practice (Mancini M., et al: Checkpoint inhibition in bladder cancer: clinical expectations, current evidence, and proposal of future strategies based on a tumor-specific immunobiological approach. Cancers 2021, 13,6016. https://doi.org/10.3390/cancers13236016)

Line 470-471: re-phrase, unclear

Please add a paragraph with strengths and limitations of the study

CONCLUSIONS

We propose a novel m7G-related scoring system and genetic signature, which seem to be able to reliably stratify the risk profile of BLCA patients, with prognostic significance. Moreover, our signature and scoring system appear to be able to indicate which patients will show a better response to immunotherapy or resistance to chemotherapy. These results could significantly improve the possibility of designing patient-oriented treatment strategies in the next future.

Author Response

Dear reviewer,

Thank you for reviewing our manuscript titled “A novel m7G-related
genes-based signature with prognostic value and predictive ability to select patients responsive to personalized treatment strategies in bladder cancer”. We have carefully studied the comments and suggestions, and then revised the manuscript accordingly. The changed and added texts in the manuscript are shown in red. We hope that the revision could be acceptable, and that our responses adequately address the comments. Should you have any questions, please contact us without hesitation.

This manuscript is interesting and offers some new data and novel findings in the field of bladder cancer biomarkers and their clinical application. However, it needs a number of changes before being accepted for publication in Cancers. The changes needed are detailed in the text below.

1 TITLE: The most important finding of this paper, which is also its main strength, is that the novel signature proposed by the Authors is able to select patients, in the preclinical setting for patient-specific therapeutical strategies in bladder cancer. This is a real novel finding. I think therefore that this fact should be clearly stated in the title of the paper, which is at the moment a bit generic and not very attention-catching. A proposed change for the title, which would make it a real hit, is:

A NOVEL m7G-RELATED GENES-BASED SIGNATURE WITH PROGNOSTIC
VALUE AND PREDICTIVE ABILITY TO SELECT PATIENTS RESPONSIVE TO
PERSONALIZED TREATMENT STRATEGIES IN BLADDER CANCER

Response: Thanks for your suggestion. We have changed the previous title “A novel signature to predict prognosis and tumor microenvironment based on m7G-related genes for bladder cancer” into “A novel m7G-related genes-based signature with prognostic value and predictive ability to select patients responsive to personalized treatment strategies in bladder cancer” in the revised manuscript (Line 2-4, Page 1).

Q2 ABSTRACT: LINE 37: In summary, our data show that m7G-related
characterization of bladder cancer patients can be of value for prognostic stratification and for patients-oriented therapeutic options, designing personalized treatment strategies in the preclinical setting.

Response: Thanks for your advice. As you suggested, we have rewritten this sentence in the revised manuscript (Line 37-40, Page 1).

Q3 INTRODUCTION:Line 48 Despite these therapies can improve the prognosis of the disease, a number of patients show a low or no response to treatments, partly because of high tumor heterogeneity. With the development of more advanced methodologies and technologies, molecular subtyping of bladder cancer cases has provided some clues for selecting upfront, before starting treatment, which patients will respond to the appropriate therapies(please cite: Roviello, G. et al. Focus on Biochemical and Clinical Predictors of Response to Immune Checkpoint Inhibitors in Metastatic Urothelial Carcinoma: Where Do We Stand? Int. J. Mol. Sci. 2020,21, 7935. https://doi.org/10.3390/ijms21217935).. Recently, novel approaches based on
bioinformatics and machine learning have attracted attention on the possibility of predicting prognosis and selecting appropriate therapies. There is still a need, however, to develop novel genetic signatures with clinical value, in order to transfer the high amount of data obtained with the bioinformatic approach to the clinical setting (reviewed with specific clinical focus in Mancini M., et al. Stem cells, Biomarkers and Genetic Profiling: approaching future challenges in Urology. Urologia J 83:4-13;2016; DOI:10.5301/uro.5000165)

Response: Thanks for your advice. As you suggested, we have rewritten this sentence in the revised manuscript (Line 48-57, Page 2).

Q4 Line 56: not“which the seventh N position”but“where the seventh N position”

Response: Thanks for your suggestion. We have modified this word in the revised manuscript (Line 58, Page 2).

Q5 Line 57: do not use “is added”, but “is complexed with”

Response: Thanks for your suggestion. We have modified this word in the revised manuscript (Line 59, , Page 2).

Q6 Line 94: More recently, with the increased understanding of m7G role in cancer, a number of studies are focusing on its role in shaping tumor microenvironment.

Response: Thanks for your advice. As you suggested, we have rewritten this sentence in the revised manuscript (Line 97-98, Page 3).

Q7 Line 117-118: erase, unnecessary repetition (from “Thus…..until the end of the sentence).

Response: Thanks for your advice. As you suggested, we have erased this sentence in the revised manuscript.

Q8 Line 123: Re-phrase. For example: Thus, we decided to investigate if this novel constructed m7G-related signature could be utilized for clinically significant patients risk stratification and also for selection, in the preclinical setting, of potentially responding patients in order to design targeted therapeutic approaches.

Response: Thanks for your advice. As you suggested, we have rewritten this sentence in the revised manuscript (Line 126-129, Page 3).

Q9 Table 1: I noticed that your group of 760 BLCA patients is not heterogeneous, being the patients mostly“high grade”and in a high stage of disease (mostly stage 3 and 4 cases). I think one of the reasons of the good correlation you are showing between the m7G signature and prognosis and therapy response is that the sample is not very heterogenous. You could have had less significant results with a more heterogenous group, including more low grade, low stage patients (which are, in the clinical real world, the majority of BLCA patients). This fact could be a limitation, but it is also a strength of this study, because the patients with high grade and high stage disease are the patients with the worst prognosis and that more often require additional therapies after or before surgery. You should discuss this point at the end of
the discussion section, talking about strengths and limitations of your paper (a
paragraph that is missing, but need to be added)

Response: Thanks for your suggestion. We have discussed this as a limitation and strength of our study in the revised manuscript (Line 499-505,  Page 18).

Q10 DISCUSSION Line 451: Erase “with a breakthrough in immunotherapy”(unnecessary). Start as: Encouraging data from clinical studies
indicated that immunotherapeutic strategies could become a major treatment strategy in BLCA. In this respect, the possibility of identifying early on, before starting treatment, which patients will respond to immunotherapy, is a major clinical need. A recent paper showed the possibility of patient-specific immunoprofiling of bladder cancer cases in order to select which patients will respond to immunotherapy, but no specific biomarkers are currently used yet in clinical practice (Mancini M., et al: Checkpoint inhibition in bladder cancer: clinical expectations, current evidence, and proposal of future strategies based on a tumor-specific immunobiological approach. Cancers 2021, 13,6016. https://doi.org/10.3390/cancers13236016)

Response: Thanks for your advice. As you suggested, we have rewritten this sentence in the revised manuscript (Line 458-464, Page 17).

Q12 Please add a paragraph with strengths and limitations of the study
Response: Thanks for your suggestion. We have added a paragraph with strengths and limitations of the study in the revised manuscript (Line 495-507, Page 18).

Q13 CONCLUSIONS We propose a novel m7G-related scoring system and genetic signature, which seem to be able to reliably stratify the risk profile of BLCA patients,with prognostic significance. Moreover, our signature and scoring system appear to be able to indicate which patients will show a better response to immunotherapy or resistance to chemotherapy. These results could significantly improve the possibility of designing patient-oriented treatment strategies in the next futur.

Response: Thanks for your advice. As you suggested, we have rewritten this sentence in the revised manuscript (Line 515-520, Page 18)

Round 2

Reviewer 4 Report

The Authors did a good job in revising the paper. There is only a minor correction to do now on the last paragraph of the Discussion, line 508, page 18 (although the clarification of the role of m7G-related modifications in the development of bladder cancer is still in a  preliminary phase ...).

For the rest, the paper is now suitable to publication in Cancers

Author Response

Dear reviewer,

Thank you for reviewing our manuscript titled “A novel m7G-related genes-based signature with prognostic value and predictive ability to select patients responsive to personalized treatment strategies in bladder cancer”. We have carefully studied the comments and suggestions, and then revised the manuscript accordingly. The changed and added texts in the manuscript are shown in green. We hope that the revision could be acceptable, and that our responses adequately address the comments. Should you have any questions, please contact us without hesitation.

The Authors did a good job in revising the paper. There is only a minor correction to do now on the last paragraph of the Discussion, line 508, page 18 (although the clarification of the role of m7G-related modifications in the development of bladder cancer is still in a preliminary phase ...).

For the rest, the paper is now suitable to publication in Cancers

Response: Thanks for your advice. As you suggested, we have corrected this sentence in the revised manuscript (Line 508-509, Page 18).